

# Tone classification of online medical services based on 1DCNN-BiLSTM

Cheng Huang[1], Peng Xie[2], Chunming Wu[1], Xiaojuan Liu[3] and Lin Zhang[1]

[1] College of Computer and Information Science, Southwest University, Beibei District, Chongqing, China
[2] School of Journalism & Communication, Chongqing University, Shapingba District, Chongqing, China
[3] School of Artificial Intelligence, Chongqing University of Technology, Liangjiang New District, Chongqing, China

## ABSTRACT

In order to improve the recognition rate of the tone classification of doctors in online medical services scenarios, we propose a model that integrates a one-dimensional convolutional neural network (1DCNN) with a bidirectional long short-term memory network (BiLSTM). Firstly, significant tone types within online medical services scenarios were identified through a survey questionnaire. Secondly, 68 features in both the time and frequency domains of doctors' tone were extracted using Librosa, serving as the initial input for the model. We utilize the 1DCNN branch to extract local features in the time and frequency domains, while the BiLSTM branch captures the global sequential features of the audio, and a feature-level fusion is performed to enhance tone classification effectiveness. When applied in online medical services scenarios, experimental results show that the model achieved an average recognition rate of 84.4% and an F1 score of 84.4%, significantly outperforming other models and effectively improving the efficiency of doctor-patient communication. Additionally, a series of ablation experiments were conducted to validate the effectiveness of the 1DCNN and BiLSTM modules and the parameter settings.

## INTRODUCTION

As living standards improve, there is an increasing focus on health. This heightened attention to health has gradually prompted a transformation in healthcare service models (*Shang & Liu, 2016*). Online medical services, with their unique convenience and efficiency, have gradually become a part of people's daily lives. Compared to traditional offline medical care, online medical services not only offer rapid consultation responses but also effectively reduce the patient load on hospitals. Under this emerging service model, the tone and manner in which doctors communicate online while answering questions have a significant impact on patients' psychological and emotional states. Research indicates that a positive communication tone can not only improve the patient's experience but may also significantly affect their treatment outcomes and satisfaction (*Liu et al., 2020*; *Wu & Lu, 2021*). Given this context, a model is proposed for automatically classifying the tone of doctors in online medical services. This enables doctors to effectively adjust their online communication strategies, thereby meeting the emotional and psychological needs

Corresponding authors
Peng Xie, xiepeng@cqu.edu.cn
Chunming Wu,
chunmingnone@163.com

of patients and ultimately enhancing the overall quality of online medical services. The study has been ethically approved by the Academic Committee of the School of Journalism at Chongqing University. Informed consent form has been provided to the participants, and written consent was obtained from them.

In recent years, traditional audio classification methods such as support vector machines (SVM), k-nearest neighbors (KNN), decision trees (DT), random forests (RF), and logistic regression (LR) have been widely used. Support vector machines (SVM) are an effective supervised learning algorithm, primarily achieving data point classification by constructing one or more hyperplanes in the feature space (*Bahatti et al., 2016*). The k-nearest neighbors (KNN) algorithm classifies based directly on the nearest samples, thereby effectively utilizing the characteristics of audio signals (*Thiruvengatanadhan, 2017*). *Pavan & Dhanalakshmi (2022)* effectively predicted the types of audio files using decision tree (DT) and random forest (RF) models. *Singh, Singh & Saluja (2024)* implemented classification of multiple emotions using a logistic regression (LR) model. However, these machine learning models have certain limitations, they typically require manual feature extraction and selection, which is not only time-consuming but may also limit classification performance due to improper feature selection. Moreover, for nonlinear and complex emotional expressions, these models may struggle to capture subtle patterns and temporal relationships in audio data. The multilayer perceptron (MLP), as a type of artificial neural network, is more effective in processing audio data compared to other machine learning models (*Karthikeyan & Mala, 2018*), yet its structure is relatively simple when compared to deep learning models.

Deep learning, with its capability to learn complex patterns, can achieve superior performance in audio classification tasks. Convolutional neural networks (CNN) and recurrent neural networks (RNN) represent two major distinct types of neural network model, *Hershey et al. (2017)* employed various CNN architectures for sound classification, specifically using models such as AlexNet (*Krizhevsky, Sutskever & Hinton, 2017*), ResNet (*He et al., 2016*), to demonstrate the effectiveness of audio classification. However, the efficiency of two-dimensional convolutional neural networks (2DCNN) in processing audio tasks is relatively low. *Abdoli, Cardinal & Koerich (2019)* proposed an end-to-end network using one-dimensional convolutional neural networks (1DCNN) for sound classification, which significantly reduced the model's training time. Although CNNs models are capable of capturing spatial features in data, they exhibit some limitations when processing time series data. Recurrent neural networks (RNN) are effective at extracting features from sequences, and long short-term memory (LSTM) is a variant of the RNN architecture. *Kanjanawattana et al. (2022)* found that LSTM performs better than CNN in emotion classification. However, unlike CNN, LSTM does not have the capability to process local features of data. *Chen & Liu (2021)* proposed a method for the cascaded fusion of CNN and bi-directional long short-term memory (BiLSTM) for audio classification, achieving better performance than using either CNN or LSTM models alone. Nevertheless, this two-stage approach increases computational complexity. To address the aforementioned challenges, we propose a model that performs feature-level fusion

of 1DCNN and BiLSTM for classifying the tone of doctors on online medical platforms. Overall, the main contributions of this article include:

1. Designed the 'questionnaire on tone types of medical professional service providers', which substantiates the rationale for categorizing the tones of doctors on online medical platforms into six distinct types.

2. Created a proprietary dataset comprising platforms 'Ding Xiang Doctor' and 'Chun Yu Doctor', and trained it on a dual-channel model constructed with CNN and BiLSTM (1DCNN-BiLSTM), effectively recognizing six tones: normal, angry, stressed, tenderness, determination, and steadiness.

3. Compared the performance of our model with other models based on evaluation metrics such as accuracy, precision, recall, F1 score, and Kappa value, and utilized ablation study to justify the appropriateness of our model's settings and parameters.

The rest of this article is organized as follows: the 'Related Work' section introduces tone classification and the process of speech feature extraction. The 'Proposed Methodology' section describes the 1DCNN-BiLSTM model that this article proposes. The 'Experiments and Results' section discusses the establishment of the dataset, the settings of model parameters, a comparison of the effects between different models, and a performance comparison of ablation study. The 'Conclusion' section summarizes the practical and theoretical significance of the proposed method, its shortcomings and areas for improvement, and future directions for application.

## Related work
## Tone classification

A person's tone often conveys their emotions and attitudes, when classifying tones, it is essential to fully consider the characteristics of each tone. *Kawade et al. (2022)* roughly categorized tones into happy, sad, angry, surprised, neutral, disgust, calm, and fear, and conducted training on the English database of the RAVDESS dataset. *Andronati et al. (2023)* also used the RAVDESS database but categorized tones into calm, happy, sad, angry, fear, surprised, and disgust. *Kanjanawattana et al. (2022)* classified tones as normal, angry, surprised, happy, and sad. the aforementioned studies share a similar issue: they merely categorize tones simplistically without thoroughly considering the appropriateness of these classifications. Given that in online medical contexts, the various tones used by doctors can significantly impact the patient, potentially affecting their health conditions severely, it is essential to classify tones accurately.

For this study, we created the "Survey on Tone Types in Specialized Medical Services" to identify the tones in professional service providers that consumers consider to be significant and important. The questionnaire design is detailed in Article S1. The importance of tone types was determined based on the impact of emotions and characteristic signals released by doctors' tones, as perceived by respondents in their medical practice experiences, on their satisfaction levels.

We recruited 261 respondents to fill out the questionnaire. After excluding 79 questionnaires with a short response time (less than 60 s), 182 valid questionnaires remained, resulting in a validity rate of 69.73%. The study has been ethically approved by
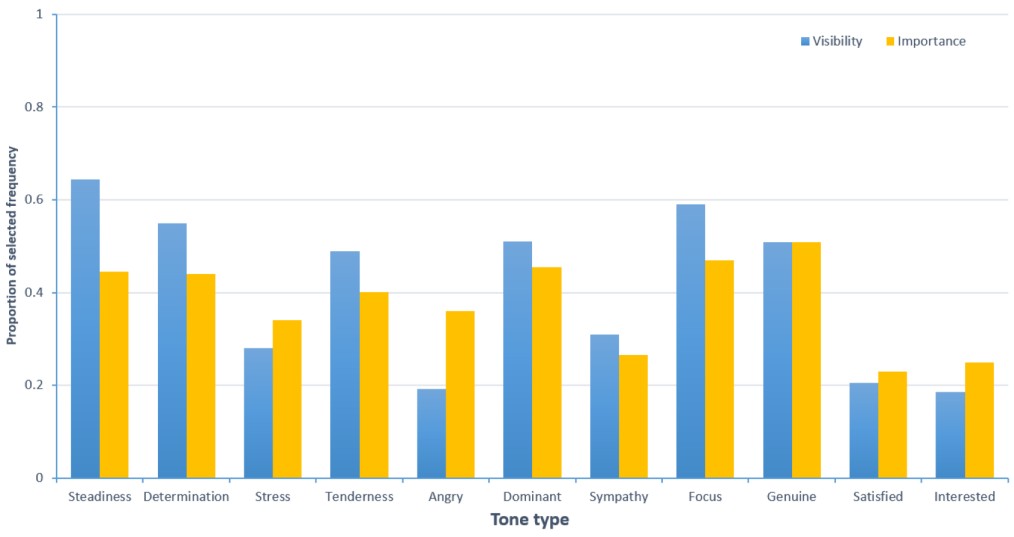

**Figure 1** **Prominence and importance of tone in specialized medical service providers.**

the Academic Committee of the School of Journalism at Chongqing University. Informed consent form has been provided to the participants, and written consent was obtained from them. As shown in Fig. 1, the tones of "determination", "steadiness", "stress", "tenderness", and "angry" are of certain importance and prominence in the context of online medical services. Notably, from the perspectives of significance and prominence, the tones of "satisfied" and "interested" are relatively low in both online medical services. Therefore, we will not measure and discuss the "satisfaction" and "interest" tone categories.

In order to refine the categorization of tone types, we based our analysis on the consistency calculated from the ratings of eight annotators on 60 audio samples, providing evidence for the reliability and validity of the voice measurements. The results, as shown in Table 1, indicate that the intraclass correlation coefficients (ICC) for sympathy, genuine, dominant, and focus are lower. Therefore, these tones will not be discussed further in this article (*Goldman, 2017*; *Wang et al., 2021*). Overall, this article identifies six categories of tone: determination, steadiness, stress, tenderness, angry, and normal.

## Feature extraction

In traditional speech and audio classification, audio signal processing is always based on the Mel-frequency cepstral coefficients (MFCC) (*Iskhakova, Wolf & Meshcheryakov, 2020*). The steps are shown in Fig. 2, which mainly include audio framing and windowing, computing the amplitude spectrum using Fourier transform, taking the logarithm of the amplitude spectrum, converting to Mel-scale frequency, and performing the discrete cosine transform. *Al-Hattab, Zaki & Shafie (2021)* utilized MFCC for environmental sound classification, while *Neili & Sundaraj (2024)* extracted MFCC features for signal classification. However, using solely MFCC features has some limitations, specifically in capturing dynamic characteristics. When analyzing the initial audio signals, there may be

**Table 1 Intraclass correlation coefficient of each tone (ICC).**

| Tone type | ICC1 | ICC1k |
|---|---|---|
| Steadiness | 0.5151[***] | 0.8947[***] |
| Determination | 0.5091[***] | 0.8924[***] |
| Stress | 0.5206[***] | 0.8968[***] |
| Tenderness | 0.5232[***] | 0.8977[***] |
| Angry | 0.5124[***] | 0.8937[***] |
| Dominant | 0.2417[***] | 0.7183[***] |
| Sympathy | 0.1739[***] | 0.6274[***] |
| Focus | 0.0589[**] | 0.3335[**] |
| Genuine | 0.0382[*] | 0.2409[*] |

**Notes.**
[***] $p < 0.01$
[**] $p < 0.05$
[*] $p < 0.1$

**Figure 2 Mel-frequency cepstral coefficients (MFCCs) extraction process.**

a loss of crucial emotional information. *Li, Shi & Wang (2019)* demonstrated the impact of other time-domain and frequency-domain features on audio, *Lesnichaia et al. (2022)* utilized features such as zero crossing rate, spectral centroid, spectral rolloff, and chroma vectors as inputs for speech classification.

Based on this, this article not only considers MFCC features but also incorporates 34 additional characteristics, including zero crossing rate, short-term energy, entropy of energy, spectral centroid, spectral spread, spectral entropy, spectral flux, spectral rolloff, and chroma vectors. Furthermore, the increments of these 34 features are calculated. For every 15-second segment of speech, 599 frames are obtained, and 68 feature values per frame are computed using Librosa (third-party Python library). To reduce the dimensionality of the input data, the feature data of 599 frames are averaged across every three consecutive frames, converting each audio into a 200x68 vector representation. The features are listed in Table 2, and their specific descriptions and calculation methods can be found in Article S4.

**Table 2  Audio features.**

| Feature | dim | description |
| --- | --- | --- |
| Zero crossing rate | 1 | The number of times the signal crosses zero |
| Short-term energy | 1 | The strength of signal energy |
| Entropy of energy | 1 | Measurement of sudden changes |
| Spectral centroid | 1 | The "center of gravity" of the spectrum |
| Spectral spread | 1 | The distribution of audio signals around the center of the spectrum |
| Spectral entropy | 1 | Characterizing the Regularity of Speech Signal Power Spectrum |
| Spectral flux | 1 | Capturing spectral flux to measure spectral changes between two consecutive frames |
| Spectral rolloff | 1 | A frequency that is lower than a specified percentage of the total spectrum energy |
| Mfccs | 13 | Mel-frequency cepstral coefficients |
| Chroma vector | 12 | Spectral energy of 12 sound poles |
| Chroma std | 1 | The standard deviation of chroma vector |
| Delta Zero Crossing Rate | 1 | Increment of Zero Crossing Rate |
| Delta Short-term energy | 1 | Increment of Short-term energy |
| Delta Entropy of Energy | 1 | Increment of Entropy of Energy |
| Delta Spectral Centroid | 1 | Increment of Spectral Centroid |
| Delta Spectral Spread | 1 | Increment of Spectral Spread |
| Delta Spectral Entropy | 1 | Increment of Spectral Entropy |
| Delta Spectral Flux | 1 | Increment of Spectral Flux |
| Delta Spectral Rolloff | 1 | Increment of Spectral Rolloff |
| Delta Mfccs | 13 | Increment of Mfccs |
| Delta Chroma Vector | 12 | Increment of Chroma Vector |
| Delta chroma std | 1 | Increment of standard deviation of chroma vector |

## PROPOSED METHODOLOGY

We propose a 1DCNN-BiLSTM model that includes one convolutional layer and one BiLSTM layer. Local features are extracted by the CNN, while global features are extracted by the BiLSTM. A specific description of the model is provided below.

### Convolutional Neural Network

2DCNN and 3DCNN are used for complex tasks such as image processing and video understanding. The data in this article consist of one-dimensional time-series signals. Considering that 1DCNN networks can effectively process audio signals (*Chowdhury & Ross, 2020*), we employ a 1DCNN to extract local features from the audio. The theoretical framework of the 1DCNN is illustrated in Fig. 3.

The initial audio features are represented as an M\*N matrix, where M denotes the dimensions and N represents the number of columns. We extract local audio features, and the convolution calculation formula is shown as Eq. (1):

$$C = f(w * x + b). \tag{1}$$

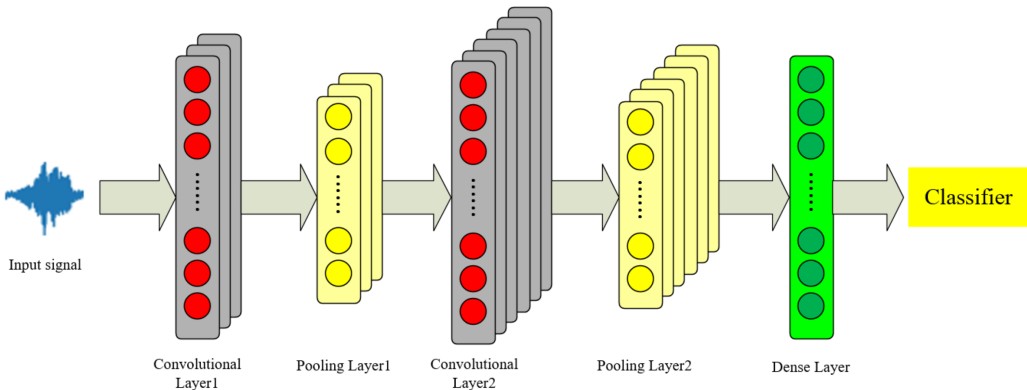

**Figure 3** **1DCNN model structure diagram.**

where $w$ is the convolution kernel, $x$ is the initial input matrix, $b$ is the bias term, $f$ is the activation function, and $C$ represents a feature map after convolution. The formula for multiple feature maps is shown in Eq. (2):

$$C_h = \left\{ C_1, C_2, \cdots, C_{\frac{m+2p-k}{s}+1} \right\}. \tag{2}$$

$C_h$ represents a series of feature maps output by the convolutional layer, $m$ is the matrix size, $P$ is the padding extension, $k$ is the size of the convolution kernel, and $s$ is the stride.

A max pooling layer is used to retain the strongest features and discard weaker ones. The formula is shown in Eq. (3):

$$C = max \left\{ C_1, C_2, \cdots, C_{\frac{m+2p-k}{s}+1} \right\} = max\{K\}. \tag{3}$$

After passing through a fully connected layer, the K vectors are concatenated into Q vectors, which serve as the input for the feature fusion layer, as shown in Eq. (4):

$$Q = max\{K_1, K_2, \ldots, K_n\}. \tag{4}$$

Equations (1) to (4) represent an example of a convolutional neural network.

## BiLSTM network

LSTM is a variant of RNN, designed to address the issues of vanishing and exploding gradients that occur during the training of long sequences (*Hochreiter & Schmidhuber, 1997*). As shown in Fig. 4, the structure of an LSTM includes an input gate, a forget gate, and an output gate.

At time $t$, the input to the LSTM is $x_t$, and the output is $h_t$. The computation of the forget gate $f_t$ at time $t$ is as Eq. (5):

$$f_t = \sigma \left( w_f \cdot \left[ h_{t-1}, x_t \right] + b_f \right). \tag{5}$$

The input gate $i_t$, determined by the previous input data and the current input, is calculated as Eq. (6):

$$i_t = \sigma \left( w_i \cdot \left[ h_{t-1}, x_t \right] + b_i \right). \tag{6}$$

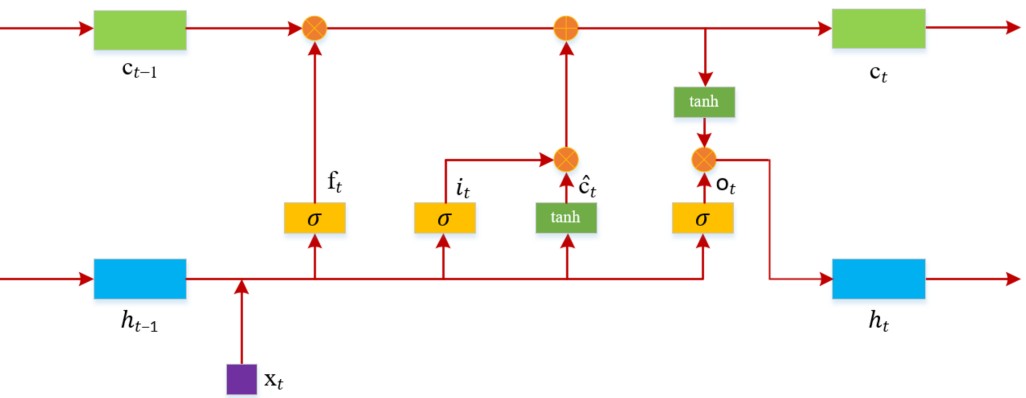

**Figure 4** LSTM model structure diagram.

At time $t$, the method for calculating the candidate values for the memory gate is as shown in Eq. (7):

$$\hat{c}_t = tanh\left(w_c \cdot \left[h_{t-1}, x_t\right] + b_c\right). \tag{7}$$

The information of the memory cell $c_t$ at time $t$ is determined based on $\hat{c}_t$ and the memory cell information from the previous moment. The computation is as shown in Eq. (8):

$$c_t = f_t \otimes c_{t-1} + i_t \otimes \hat{c}_t. \tag{8}$$

The computation of the output gate $o_t$ is as shown in Eq. (9):

$$o_t = \sigma\left(w_o \cdot \left[h_{t-1}, x_t\right] + b_o\right). \tag{9}$$

The output value $h_t$ at time $t$ is calculated as shown in Eq. (10):

$$h_t = o_t \otimes tanh(c_t). \tag{10}$$

In the equations above, $w$ represents the weights, $b$ represents the bias, $\sigma$ is the sigmoid function, and $\otimes$ signifies the dot product of vectors.

LSTM can only learn information at the current moment, whereas audio information is typically interconnected. Therefore, BiLSTM is used to simultaneously learn contextual information and capture global features (*Ibrahim, Badran & Hussien, 2022*). Its structure, as illustrated in Fig. 5, includes an additional backward layer on top of the forward layer of the LSTM model, enabling comprehensive consideration of contextual information by concatenating the forward and backward hidden layer vectors.

## PROPOSED MODEL

Combining the advantages of 1DCNN and BiLSTM mentioned above, we integrate both networks for feature-level fusion to construct the 1DCNN-BiLSTM model. The audio signals are input into the model, and the final classification results are output by the fully connected layer. Specific model framework and parameters can be found in Figs. 6 and 7.

From Figs. 6 and 7, it is evident that in the 1DCNN branch, the input layer accepts dimensions of (None, 200, 68), where None represents the batch size, 200 represents the

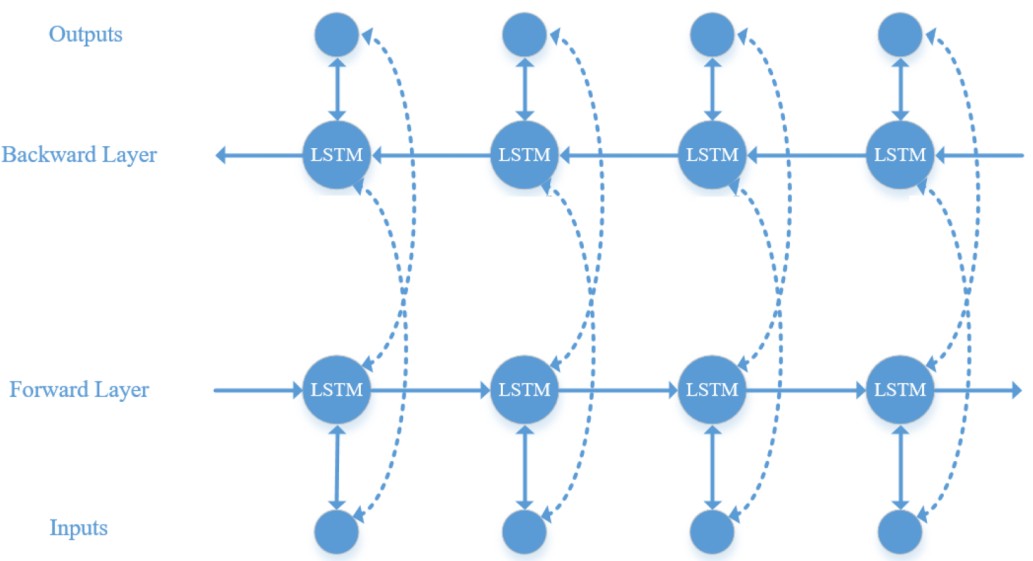

**Figure 5** BiLSTM model structure diagram.

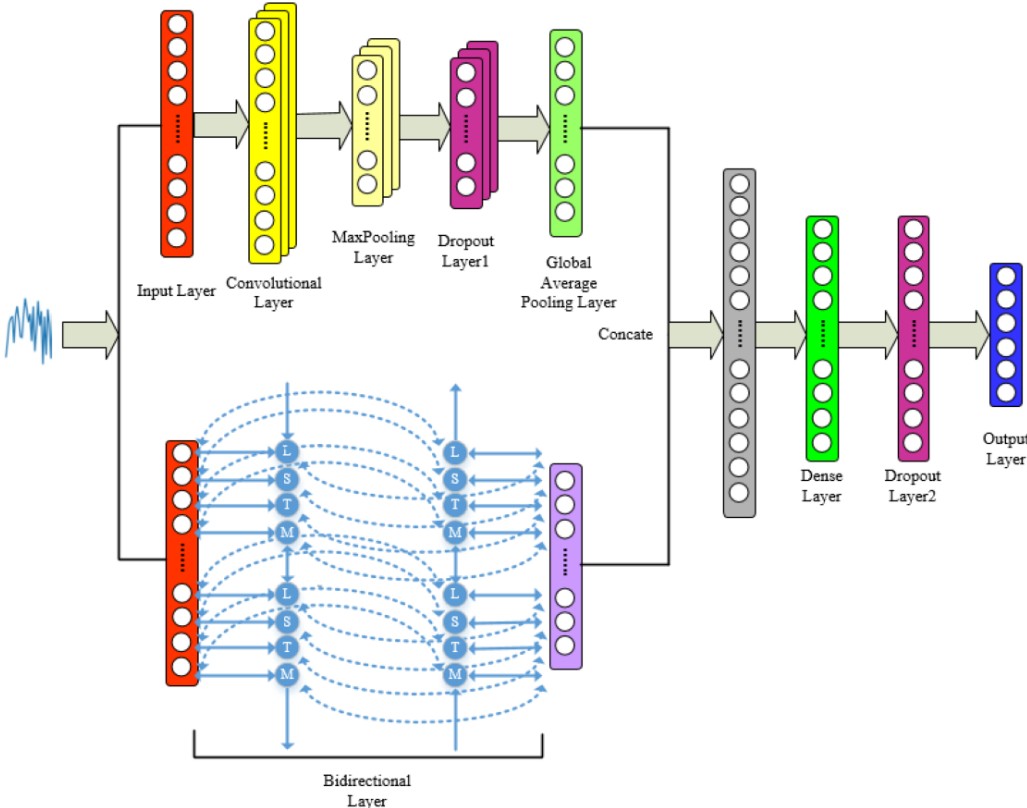

**Figure 6** Proposed model for tone recognition.

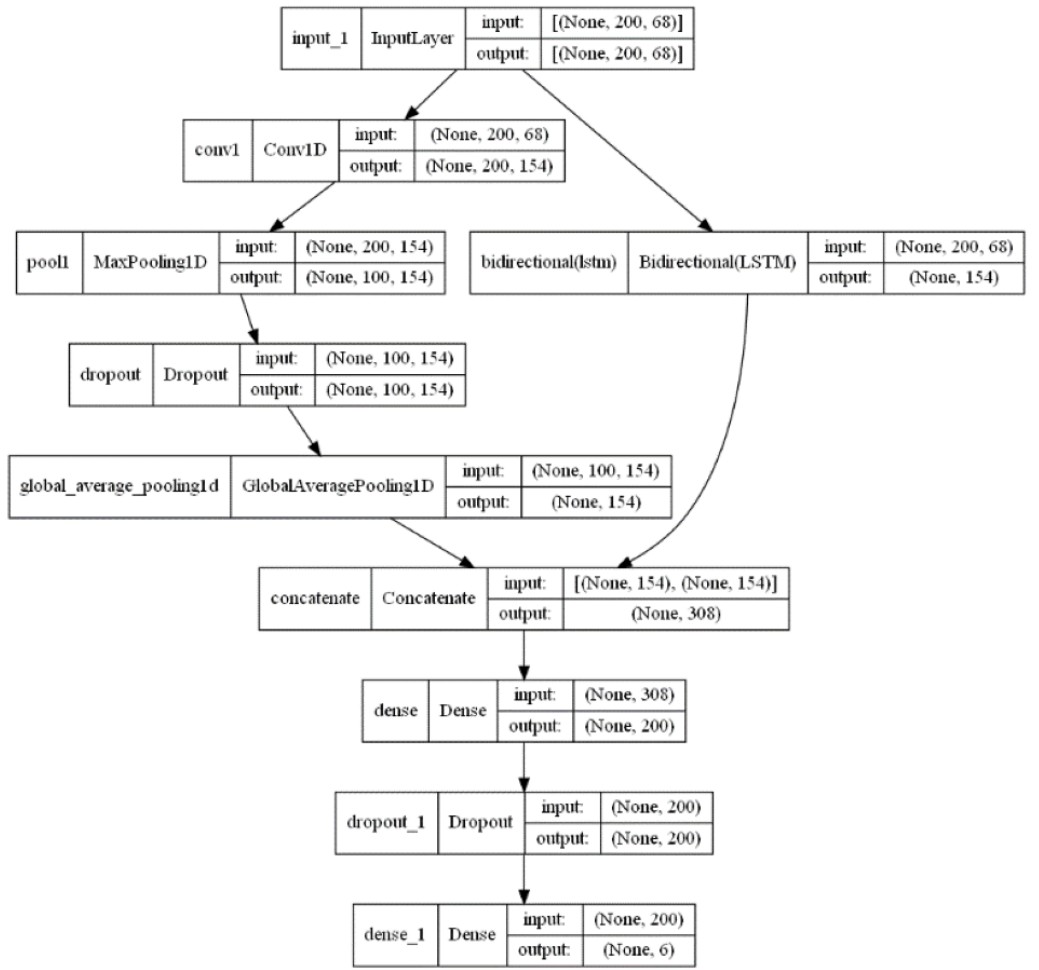

**Figure 7   1DCNN-BiLSTM model structure diagram.**

length of the time series, and 68 represents the number of audio features. This is followed by a convolution layer, which outputs dimensions of (None, 200, 154). The data then enters a max pooling layer, reducing dimensions to (None, 100, 154). A global average pooling layer averages each feature map, resulting in dimensions of (None, 154). In the BiLSTM branch, the output layer changes the dimensions to (None, 154), which are then concatenated with the 1DCNN branch, resulting in dimensions of (None, 308). This is further processed by a dense layer, adjusting dimensions to (None, 200), and finally, through another dense layer, the output dimensions are set to (None, 6), corresponding to six categories of tones.

## EXPERIMENTS AND RESULTS

### Dataset

Public datasets for online medical scenarios are rare and typically lack emotional labels. Therefore, the dataset used in this article is custom-built. "Dingxiang Doctor" (https://dxy.com) and "Chunyu Doctor" (https://www.chunyuyisheng.com) are two major

**Table 3 Experimental data set.**

| Tone | Sample size | Classification standard |
|---|---|---|
| Determination | 261 | Sounds persistent and serious |
| Steadiness | 260 | Sounds calm, cautious or self-controlled; |
| Stress | 249 | Sounds anxious and nervous |
| Angry | 260 | Sounds angry and annoyed |
| Tenderness | 253 | Sounds gentle |
| Normal | 313 | Sounds ordinary |

online medical platforms. From their databases, we randomly selected all recording logs of 1,000 doctors. From each doctor's set of recording logs, we randomly chose three question IDs and selected voice recordings longer than 15 s from each question, ultimately obtaining 3,000 raw audio samples. We then applied a low-pass filter at 450 Hz and 60dB per octave to fuzzify the original audio signals. The 3,000 audio samples were randomly divided into two parts, one part included 60 audio samples for the 'Tone Classification' section, and the other part consisting of 2,940 samples underwent manual listening and judgment. From this, we selected 300 distinct recordings each of determination, steadiness, stress, tenderness, angry, and normal (without any specific) tones, totaling 1,800 samples.

Then we divided the 1800 voice recordings into 60 tasks, which were annotated by labelers based on the audio labeling task (see Article S2). Following the verification of the validity of the annotations, we obtained 1596 effective standard voice samples as the dataset. As shown in Table 3, this dataset includes 261 "determination" tone audios, 250 "steadiness" tone audios, 249 "stress" tone audios, 260 "angry" tone audios, 253 "tenderness" tone audios, and 323 "normal" tone audios. Here, "normal" tone recordings are those that did not exceed a score of 3 (average) in any of the aforementioned five tonal categories. The specific processing procedure is depicted in Fig. 8.

## Parameter settings

As shown in Table 4, we set the number of convolution kernels in the convolutional layer of the 1DCNN branch in our model to 5, the number of feature channels to 154, the stride to 1, and the activation function to ReLU, with all dropout rates set at 0.2. In the BiLSTM branch, the number of hidden neurons is set to 77. The dataset is randomly divided into a training set and a test set in an 8:2 ratio, with a batch size of 64. The loss function is cross-entropy loss, the optimizer is Adam (*Kingma & Ba, 2014*), and the initial learning rate is set at 0.01, dynamically adjusted at a ratio of 1/10 based on learning progress. To prevent overfitting, early stopping is implemented when the validation loss remains unchanged for 15 epochs. Our model was implemented using the Tensorflow and Keras frameworks in a Python3 environment.

## Performance evaluation

This article is a multiclass classification task. We treat multiclass classification as multiple binary classification tasks, with each task yielding metrics such as accuracy, recall, and F1 score. The formulas for these metrics are presented in Eqs.(11) to (14):

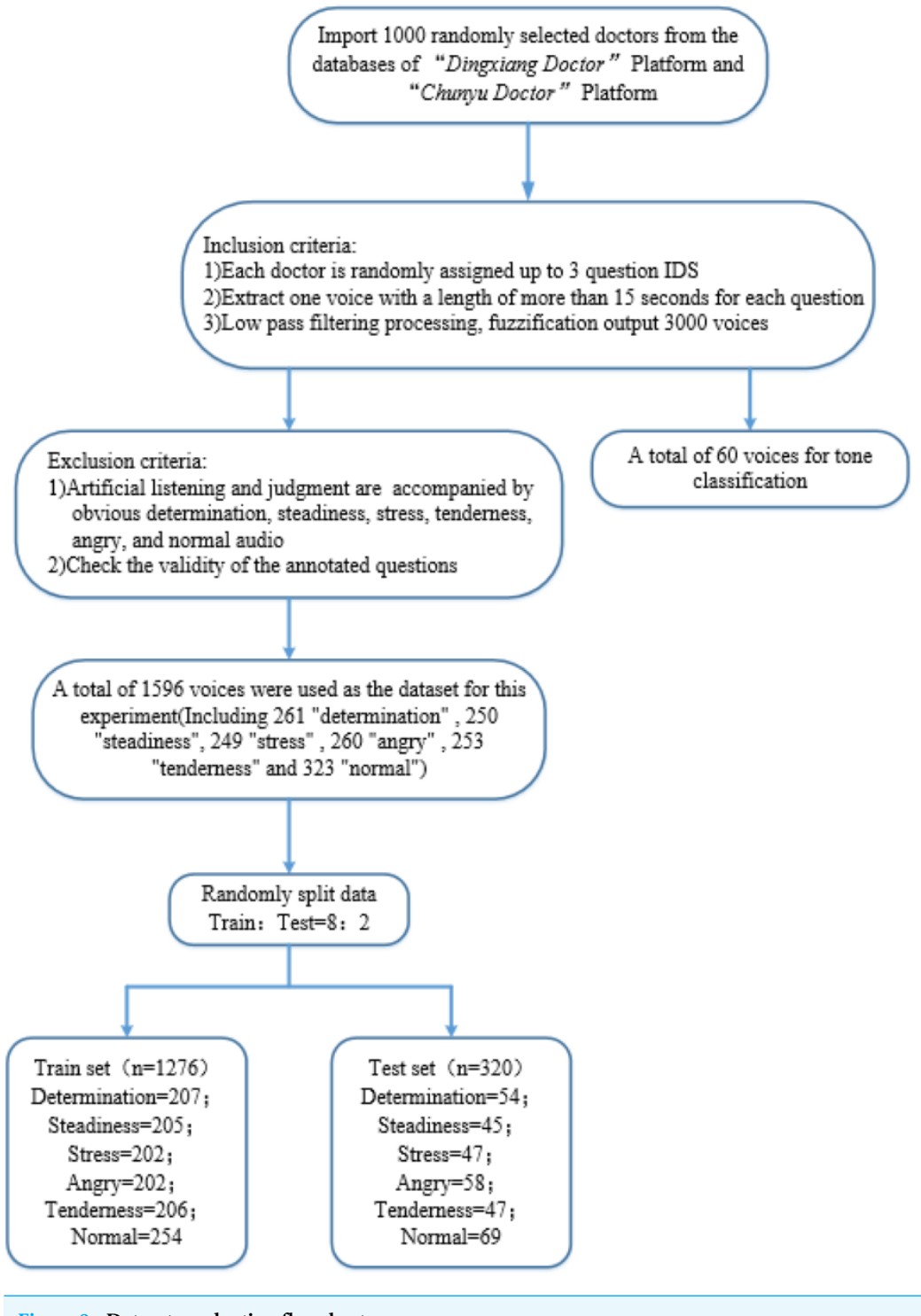

**Figure 8** Dataset production flowchart.

**Table 4  Hyperparameter settings.**

| Hyperparameter | Value |
|---|---|
| Filter size for 1DCNN | 154 |
| Kernel size for 1DCNN | 5 |
| Padding | Same |
| Unit number for BiLSTM | 77 |
| Dense layer neurons | 200 |
| Optimizer | Adam |
| Batch size | 64 |
| Epoch number | 200 |
| Activation | ReLU |

$$\text{Acc} = \frac{TP + TN}{TP + TN + FP + FN} \tag{11}$$

$$\text{Precision} = \frac{TP}{TP + FP} \tag{12}$$

$$\text{Recall} = \frac{TP}{TP + FN} \tag{13}$$

$$F1 = \frac{2 * \text{Precision} * \text{Recall}}{\text{Precision} + \text{Recall}} \tag{14}$$

where TP, FP, TN, and FN refer to the number of true positive, false positive, true negative, and false negative predictions, respectively.

Macro averaging is calculated as the arithmetic mean for each category. We apply macro averaging to the aforementioned metrics to assess and compare with other models. Besides these metrics, the Kappa coefficient is also commonly used to evaluate the accuracy of multiclass models (*Cohen, 1960*), with the specific calculation formulas provided in Eqs. (15) and (16):

$$\text{kappa} = \frac{p_0 - p_e}{1 - p_e} \tag{15}$$

$$p_e = \frac{\sum_{j=1}^{k} a_j * b_j}{n * n}. \tag{16}$$

Where $n$ represents the total number of samples, $p_0$ represents the number of samples the raters agree on and is divided by the total number of samples, $a_j$ denotes the number of actual samples in class $j$, and $b_j$ represents the number of samples predicted to be in class $j$.

**Table 5** Performance comparison of different models on the test set.

| Methods | Type | ACC | Precision | Recall | F1 | Kappa |
| --- | --- | --- | --- | --- | --- | --- |
| SVM | ML | 0.516 | 0.513 | 0.538 | 0.518 | 0.421 |
| KNN | ML | 0.431 | 0.455 | 0.439 | 0.440 | 0.317 |
| DT | ML | 0.363 | 0.364 | 0.368 | 0.364 | 0.234 |
| RF | ML | 0.488 | 0.496 | 0.496 | 0.493 | 0.383 |
| LR | ML | 0.547 | 0.548 | 0.564 | 0.545 | 0.458 |
| MLP | ANN | 0.688 | 0.691 | 0.684 | 0.685 | 0.623 |
| AlexNet | CNN | 0.428 | 0.482 | 0.429 | 0.437 | 0.307 |
| ResNet | CNN | 0.353 | 0.377 | 0.368 | 0.347 | 0.225 |
| LSTM | RNN | 0.506 | 0.503 | 0.518 | 0.497 | 0.409 |
| CNN+LSTM | Hybird | 0.534 | 0.577 | 0.527 | 0.528 | 0.435 |
| CNN+SVM | Hybird | 0.603 | 0.602 | 0.607 | 0.603 | 0.523 |
| CNN+LR | Hybird | 0.588 | 0.590 | 0.594 | 0.591 | 0.504 |
| LSTM+SVM | Hybird | 0.550 | 0.550 | 0.557 | 0.545 | 0.461 |
| LSTM+RF | Hybird | 0.547 | 0.547 | 0.551 | 0.547 | 0.455 |
| 1DCNN-BiLSTM | Hybird | 0.844 | 0.847 | 0.843 | 0.844 | 0.812 |

# RESULTS

## Classification performance

As shown in Table 5, the model proposed in this article demonstrated the highest accuracy (the average recognition rate for six tones), precision, recall, F1 score, and Kappa coefficient on the test set, achieving 84.4%, 84.7%, 84.3%, 84.4%, and 81.2%, respectively, and significantly exceeds the performance of other models. It is notable that the accuracies of five other machine learning models on the test set were relatively low, at 51.6%, 43.1%, 36.3%, 48.8%, and 54.7%, respectively. The MLP of the artificial neural networks recorded an accuracy of 68.8%, which is 15.6% lower than our model, and an F1 score of 68.5%, which is 15.9% lower than our model. The deep learning models AlexNet and ResNet had accuracies of 42.8% and 35.3%, respectively, with other performance indicators also being low, indicating that complex CNN models do not effectively extract the global sequential features of audio for tone classification. The LSTM model had an accuracy of only 50.6%, suggesting it struggles to efficiently extract local audio features. Moreover, several hybrid models also showed significant performance gaps compared to the model developed in this article, with CNN+LSTM, CNN+SVM, and CNN+KNN achieving accuracies of 51.6%, 60.3%, and 58.8% respectively, while LSTM+SVM and LSTM+KNN scored 54.1% and 51.6%.

As shown in Fig. 9, the model presented in this article demonstrates relatively high recognition rates across various tones, with an 88.4% recognition rate for the 'normal' tone. In contrast, the recognition rates for five other machine learning models are notably

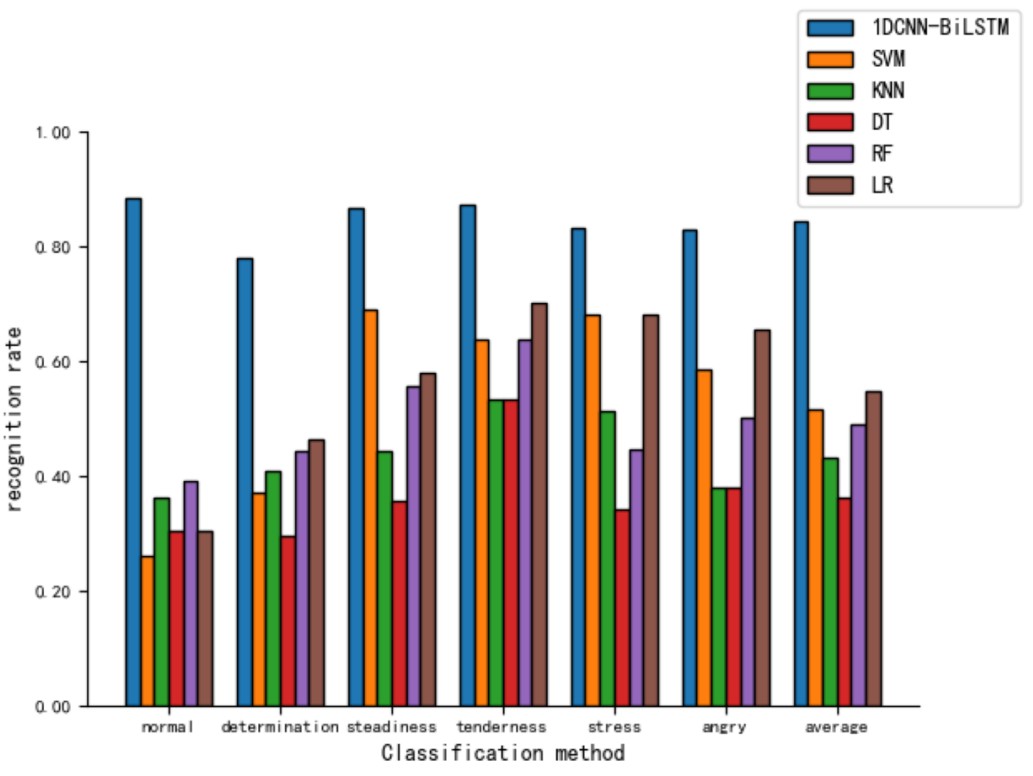

**Figure 9** Comparison of tone recognition rates on the test set between different machine learning models and the 1DCNN-BiLSTM.

lower, with the LR model achieving a 65.5% recognition rate for 'angry' and the lowest being the SVM model's recognition of 'normal' at merely 26.1%.

Figure 10 shows that the MLP recognizes different tones with rates ranging from 57% to 79%. However, the CNN models, AlexNet and ResNet, display poor recognition across all tones, particularly with ResNet's recognition ability for the 'normal' tone at only 5.8%.

Figure 11 reveals that various hybrid models perform moderately in tone recognition. In summary, the 1DCNN-BiLSTM combination model developed in this article exhibits the best performance in tone classification tasks, demonstrating high applicability.

## Confusion matrix

The confusion matrix provides a clear visualization of the recognition rates for various tones, where each row represents the actual tone label and each column represents the predicted tone label. Diagonal elements indicate correct classifications, while off-diagonal elements represent misclassifications. According to Figs. 12, 13 and 14, the recognition rates for different tones by the model presented in this article range between 78% and 89%, demonstrating high consistency between predicted and actual labels. Other models show limited capabilities in recognizing various tones. The MLP model exhibits a higher recognition rate for 'normal', but a lower rate for 'determination'. AlexNet shows lower accuracy for 'determination' and 'stress', but performs slightly better for 'steadiness' and

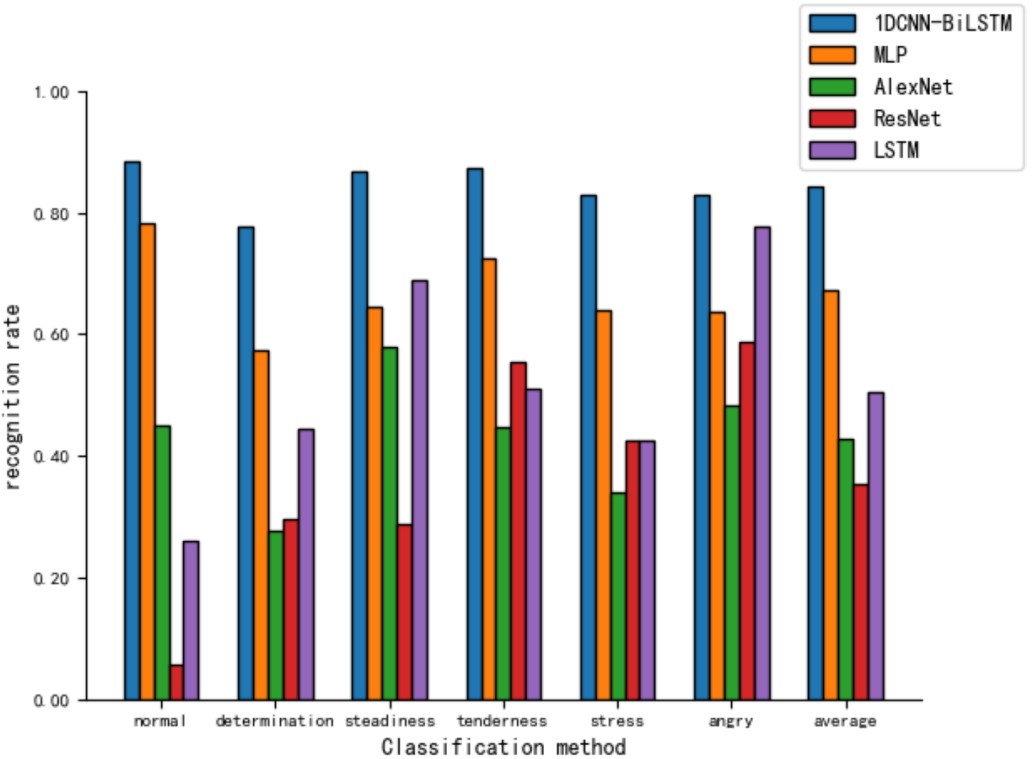

**Figure 10** **Comparison of tone recognition rates on the test set between different neural network models and the 1DCNN-BiLSTM.**

'angry'. ResNet achieves recognition rates above 50% only for 'tenderness' and 'angry', with overall low performance. The classification effectiveness of LSTM is uneven, with the highest rate for 'angry' at 77.6% and the lowest for 'normal' at only 26.1%, indicating that LSTM frequently misclassifies 'normal' as other tones. For hybrid models, the recognition rates for most tones are concentrated between 50% and 70%, the recognition results are not significant. Overall, the model proposed in this article exhibits the highest recognition rates for various tones.

## Ablation study

To validate the effectiveness of the architecture proposed in this article, two types of cascaded fusion approaches are compared with our model. Figure 15 displays the three methods of CNN and BiLSTM integration.

As shown in Table 6, it is evident that Model-I and Model-II require more parameter computations compared to our model, yet their accuracies are only 50.3% and 50.9%, respectively. The feature-level fusion approach proposed in this article not only reduces the number of parameter computations but also achieves a higher accuracy of 84.4% compared to the other two cascaded fusion methods.

Subsequently, we demonstrated the effectiveness of the modules by using 1DCNN and BiLSTM as two basic modules, respectively named Base1 and Base2. We constructed
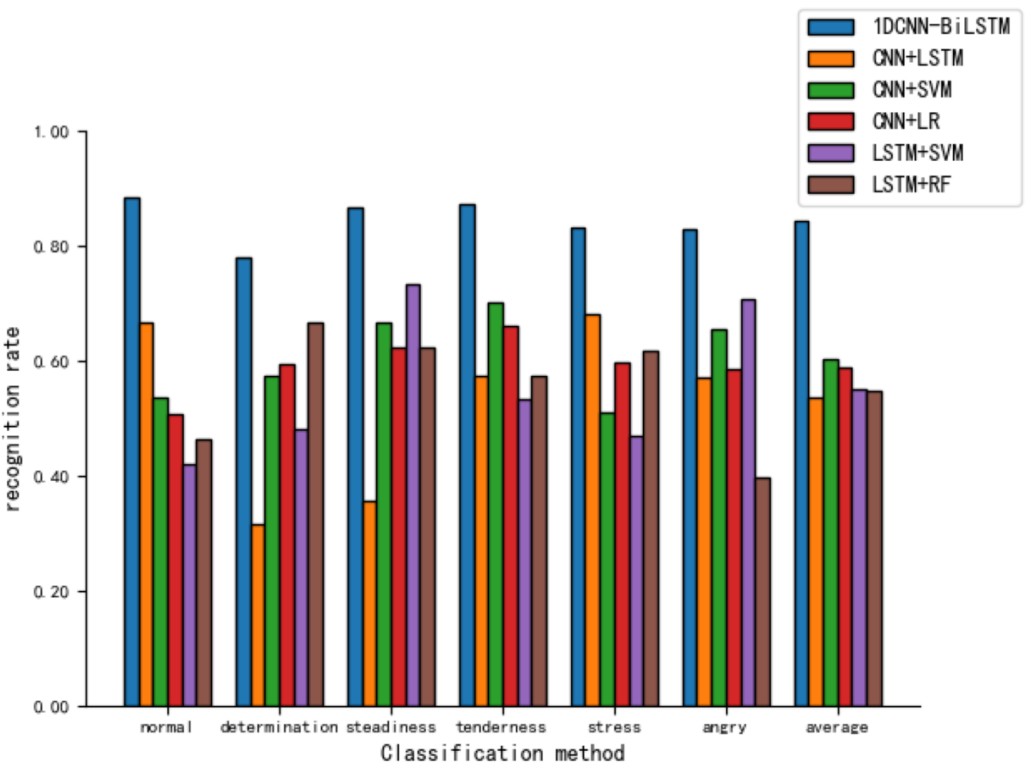

**Figure 11** **Comparison of tone recognition rates on the test set between different hybird models and the 1DCNN-BiLSTM.**

**Table 6** **Ablation study on the impact of the different model architectures.**

| Methods | Params (M) | ACC | F1 |
|---|---|---|---|
| Model I | 0.87 | 0.593 | 0.549 |
| Model II | 0.96 | 0.588 | 0.578 |
| Model III (ours) | 0.78 | 0.844 | 0.844 |

different model combinations, and the results are shown in Table 7. It can be seen that using only Base1 or Base2 models yields poor results on our custom dataset, indicating that standalone 1DCNN and BiLSTM modules are not effective in extracting features from audio data by themselves. However, when these modules are fused together, both local features and global sequential features are extracted, resulting in a significant improvement in classification performance.

We also verified the impact of different dropout rates on the model's performance. Specifically, we set the dropout rates to 0.1 and 0.2, with the specific results shown in Table 8. It can be observed that when the dropout rate is set to 0.2, the model achieves better performance. In contrast, setting the dropout rate to 0.1 results in decreased generalization capability, leading to poorer metrics across all categories on the test set.

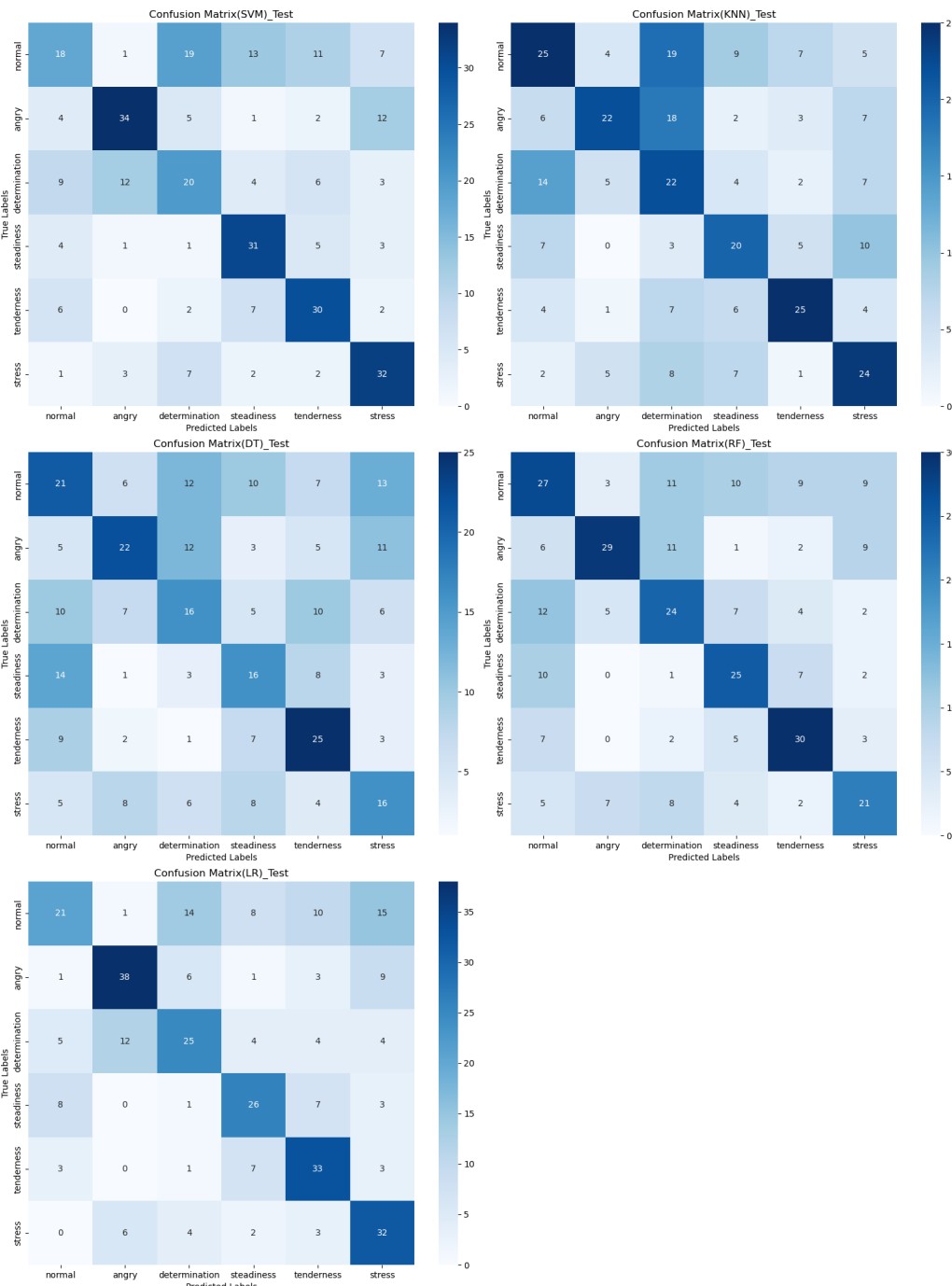

**Figure 12** Confusion matrix visualization of different machine learning models on the test set.

Furthermore, we set the number of network layers in the BiLSTM module to 1 and 2, with the results shown in Table 9. When the number of layers in the BiLSTM module increased from 1 to 2, the dual-layer BiLSTM led to excessive transmission and overprocessing of information. This caused the model to lose critical audio features or sequence information

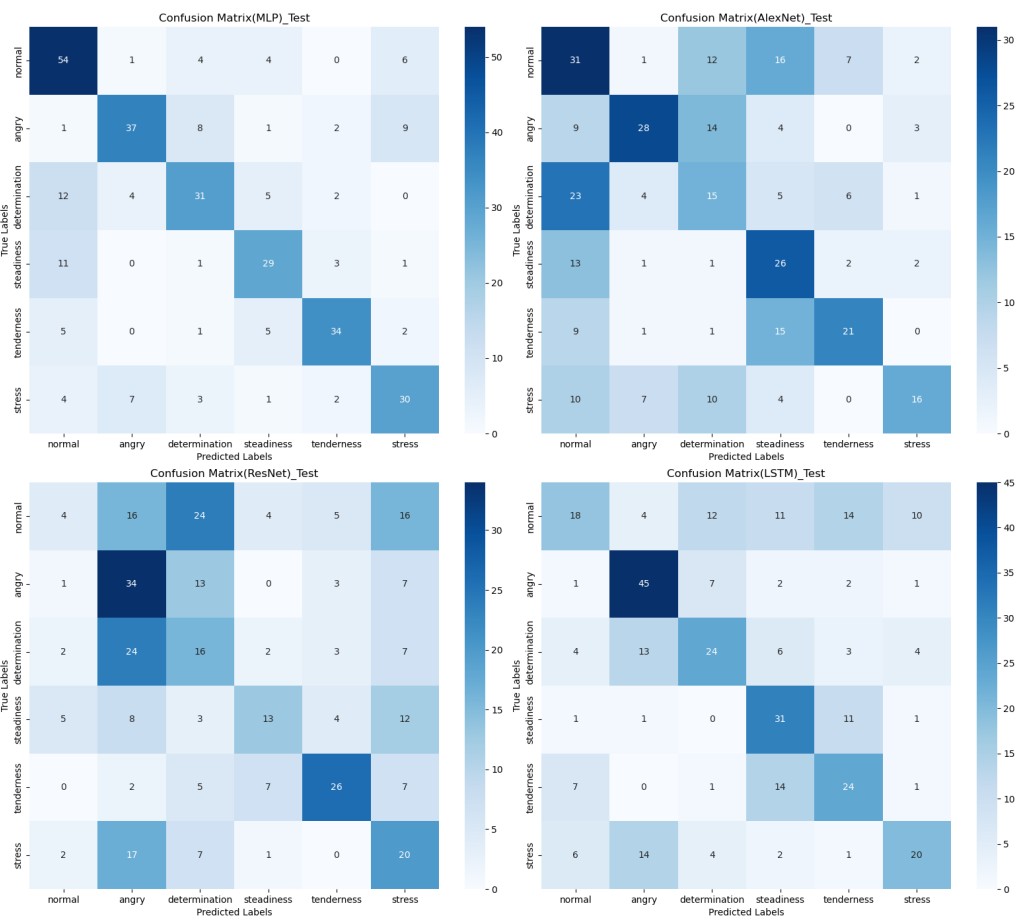

**Figure 13   Confusion matrix visualization of different neural network models on the test set.**

**Table 7   Ablation study on the impact of the different module combinations.**

| Tone Type | Tone recognition rate | | | | | |
|---|---|---|---|---|---|---|
| | Train | | | Test | | |
| | Base1 | Base2 | Base1+Base2 (ours) | Base1 | Base2 | Base1+Base2 (ours) |
| Normal | 0.539 | 0.460 | 0.886 | 0.565 | 0.420 | 0.884 |
| Determination | 0.599 | 0.604 | 0.816 | 0.556 | 0.611 | 0.778 |
| Steadiness | 0.551 | 0.434 | 0.853 | 0.511 | 0.378 | 0.867 |
| Tenderness | 0.592 | 0.573 | 0.811 | 0.617 | 0.596 | 0.872 |
| Stress | 0.371 | 0.525 | 0.832 | 0.319 | 0.489 | 0.830 |
| Angry | 0.436 | 0.505 | 0.866 | 0.293 | 0.448 | 0.828 |
| Average | 0.516 | 0.515 | 0.846 | 0.478 | 0.488 | 0.844 |

during the learning process, thereby negatively impacting the accuracy of the classification task.

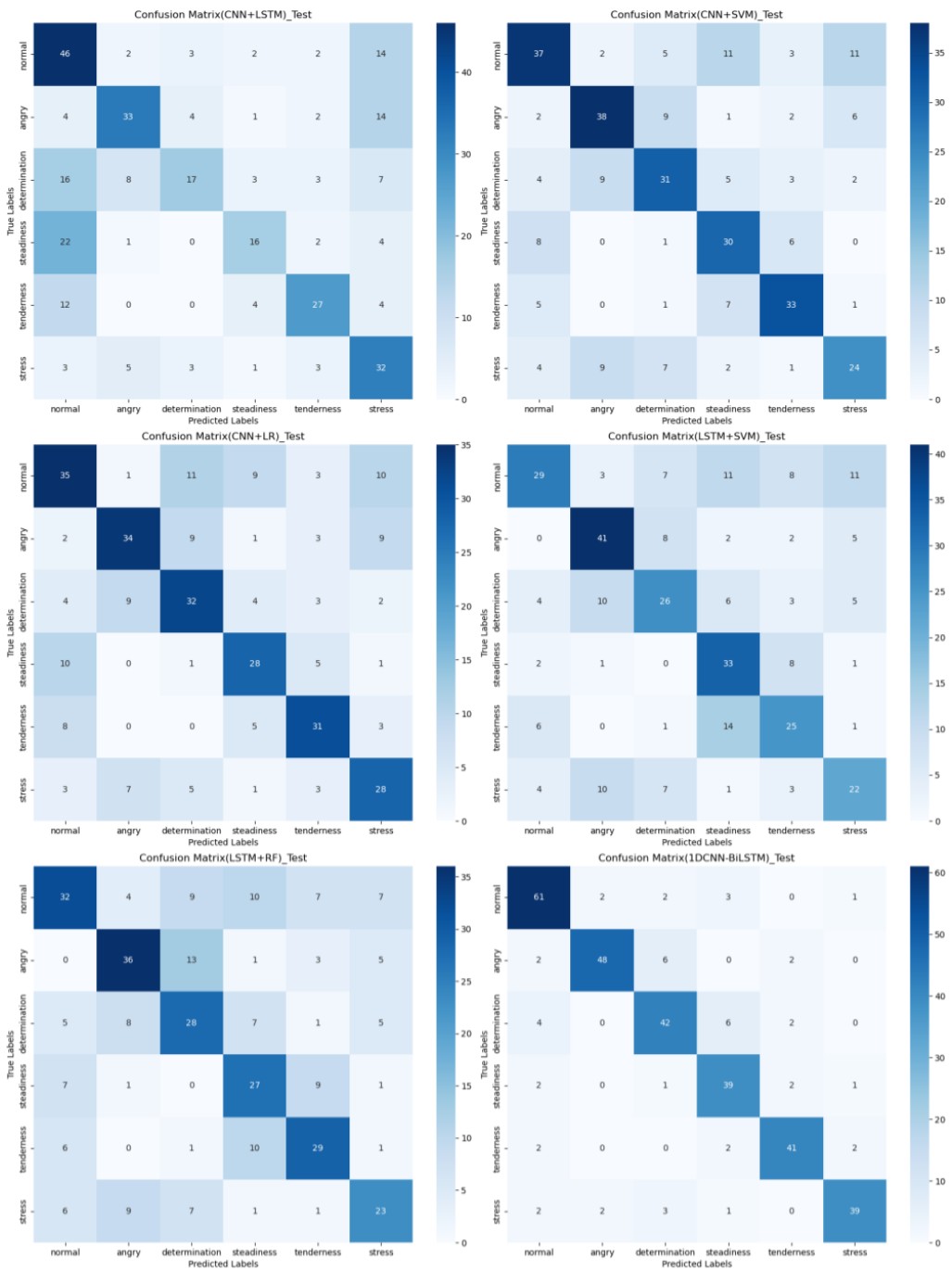

**Figure 14  Confusion matrix visualization of different hybird models on the test set.**

Finally, we also set the batch size to 64, 128, and 256, with the specific results shown in Table 10. As the batch size increased from 64 to 128, and then from 128 to 256, the increase in batch size did not enhance the model's performance. Instead, it prevented the model

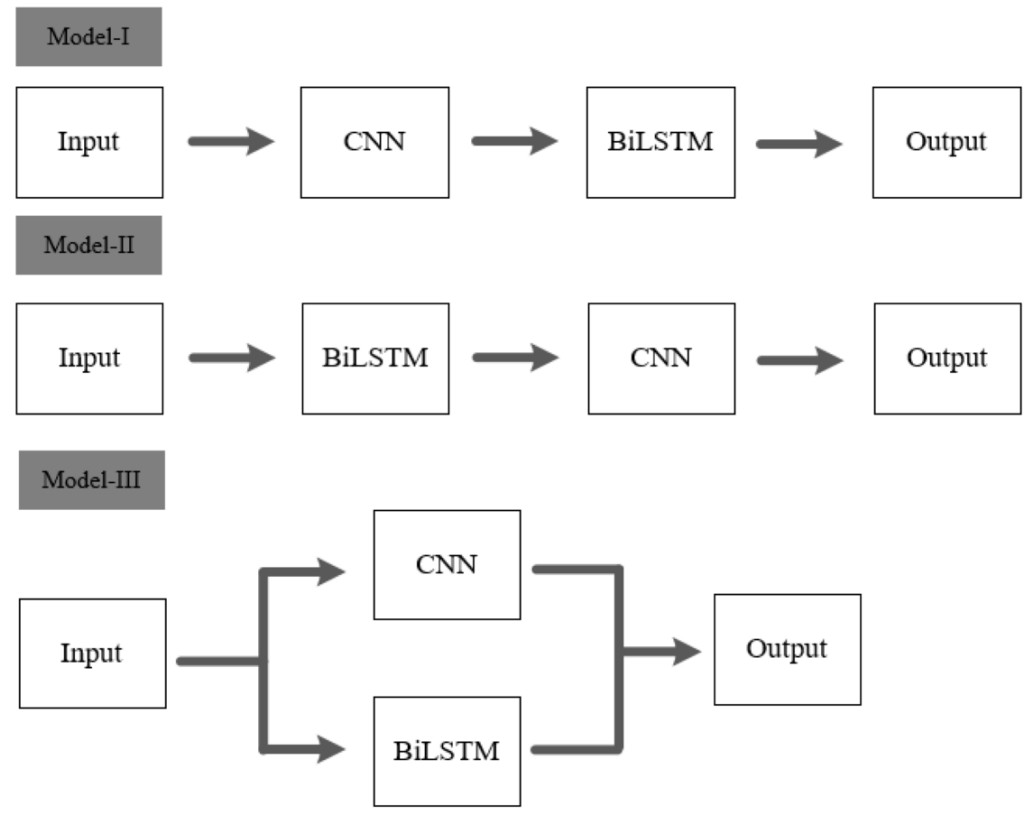

**Figure 15** **Different model architectures.** Model I and Model II are two different cascaded fusion networks, Model III is the feature-level fusion network proposed in this article.

**Table 8** **Ablation study on the impact of the dropout rate.**

| Dropout rate | ACC | Precision | Recall | F1 | Kappa |
|---|---|---|---|---|---|
| 0.1 | 0.372 | 0.348 | 0.359 | 0.315 | 0.231 |
| 0.2 (ours) | 0.844 | 0.847 | 0.843 | 0.844 | 0.812 |

**Table 9** **Ablation study on the impact of the BiLSTM layer.**

| BiLSTM layer | ACC | Precision | Recall | F1 | Kappa |
|---|---|---|---|---|---|
| 1 (ours) | 0.844 | 0.847 | 0.843 | 0.844 | 0.812 |
| 2 | 0.406 | 0.472 | 0.392 | 0.363 | 0.275 |

from effectively learning individual sample characteristics, leading to a significant decline in accuracy and other performance metrics.

## CONCLUSION

We based on deep learning theory and building upon the characteristics of one-dimensional convolutional neural networks and long short-term memory networks, successfully

**Table 10  Ablation study on the impact of the batch size.**

| Batch size | ACC | Precision | Recall | F1 | Kappa |
|---|---|---|---|---|---|
| 64 (ours) | 0.844 | 0.847 | 0.843 | 0.844 | 0.812 |
| 128 | 0.538 | 0.547 | 0.544 | 0.538 | 0.444 |
| 256 | 0.416 | 0.538 | 0.415 | 0.414 | 0.293 |

constructs an online medical services tone classification model using 1DCNN-BiLSTM. The specific categories of tone were determined through surveys, and the model classifies six key tones of interest on a custom dataset. Compared to other machine learning, deep learning, and hybrid models, our model achieved the best performance, with an average accuracy of 84.4%. An ablation study demonstrates the advantages of feature-level fusion of 1DCNN with BiLSTM over traditional cascaded fusion, as well as the rationality of the parameter settings. Although the proposed 1DCNN-BiLSTM model has shown good results in tone classification within an online medical context, this study acknowledges certain limitations in the findings.

Firstly, our parameter tuning focused only on the settings of dropout rate, batch size, and the number of network layers, without addressing other hyperparameters such as the optimizer, activation function, and loss function. In the next steps, we will explore these additional hyperparameters to further optimize the model.

Secondly, the proposed method was only trained and tested in the context of medical service scenarios. It has not been applied to other contexts such as shopping or education, so future research will investigate the effectiveness of the 1DCNN-BiLSTM model in these additional scenarios.

Lastly, the model has not been validated and tested on large-scale public datasets; therefore, we plan to further validate the model's effectiveness using public datasets in subsequent steps.

### Funding
This work was supported by the Scientific and Technological Research Program of Chongqing Municipal Education Commission (No. KJQN202301162), the Scientific Research Foundation of Chongqing University of Technology (No. 0121230235), the Chongqing Language and Writing Research Funds (No. yyk23208), and the Fundamental Research Funds for the Central Universities (No. 2023CDSKXYXW008). The funders had no role in study design, data collection and analysis, decision to publish, or preparation of the manuscript.

### Grant Disclosures
The following grant information was disclosed by the authors:
The Scientific and Technological Research Program of Chongqing Municipal Education Commission: KJQN202301162.

The Scientific Research Foundation of Chongqing University of Technology: 0121230235.
The Chongqing Language and Writing Research Funds: yyk23208.
The Fundamental Research Funds for the Central Universities: 2023CDSKXYYXW008.

## Competing Interests

The authors declare that there are no competing interests.

## Author Contributions

- Cheng Huang conceived and designed the experiments, performed the experiments, analyzed the data, performed the computation work, prepared figures and/or tables, authored or reviewed drafts of the article, and approved the final draft.
- Peng Xie conceived and designed the experiments, performed the computation work, authored or reviewed drafts of the article, and approved the final draft.
- Chunming Wu performed the experiments, authored or reviewed drafts of the article, and approved the final draft.
- Xiaojuan Liu analyzed the data, authored or reviewed drafts of the article, and approved the final draft.
- Lin Zhang performed the computation work, authored or reviewed drafts of the article, and approved the final draft.

## Ethics

The following information was supplied relating to ethical approvals (*i.e.*, approving body and any reference numbers):

The Academic Committee of the School of Journalism & Communication at Chongqing University has granted ethical approval for the conduct of this study.

## Data Availability

The executable code is available in the Supplementary Files.

The third-party data is available at:

- Dingxiang Doctor, https://dxy.com

- Chunyu Doctor, https://www.chunyuyisheng.com.

The combined dataset from the third party datasets are available at figshare: Huang, Cheng (2024). Raw Data.zip. figshare. Dataset. https://doi.org/10.6084/m9.figshare.25013849.v2.

## Supplemental Information

Supplemental information for this article can be found online at http://dx.doi.org/10.7717/peerj-cs.2325#supplemental-information.

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
