# Peer review of "Tone classification of online medical services based on 1DCNN-BiLSTM"

_PeerJ Computer Science, doi:10.7717/peerj-cs.2325_

## Round 0.1 · original submission · Major Revisions

Dear Authors,
Thank you for submitting your manuscript titled " Research on tone classification and recognition of online medical specialized services based on 1DCNN-BiLSTM" to PeerJ Computer Science. Based on the reviews received, major revisions are necessary before further consideration for publication. Below is a summary of the reviewers' comments that require your attention:

Reviewer 1:
1. The title is lengthy, and the abstract is incomplete.
2. The workflow and data provided are insufficient.
3. The methodology and results need clearer explanations.
4. Significant grammar and clarity issues.

Reviewer 2:
1. The abstract lacks detail about the tone of online medical services.
2. The use of CNN with LSTM increases complexity without clear justification.
3. The introduction and literature review need significant improvement.
4. The equations and results sections require better formatting and explanation.
5. A table showing dataset classes and features is needed.

Reviewer 3:
1. The experiments lack innovation and only compare with some machine learning algorithms.
2. The theoretical approach needs a more practical comparison with recent algorithms.
3. References should include more recent studies.

Reviewer 4:
1. The introduction must state the problem's challenge based on recent methods.
2. The methodology and enhancements to algorithms need clearer presentation and justification.
3. The experimental design and selection of baseline methods require better explanation.
4. The results and performance of the proposed method are not adequately discussed.
5. Most references are outdated and need updating.

Action Items for Major Revisions:
1. Shorten and refine the title and abstract.
2. Provide a comprehensive workflow and sufficient data details.
3. Clarify the methodology, parameter selection, and experimental design.
4. Improve grammar, clarity, and overall writing structure.
5. Justify the use of CNN with LSTM and the selection of baseline methods.
6. Include a detailed introduction, literature review, and relevant comparisons.
7. Format equations properly and provide detailed explanations of results.
8. Update references with recent studies and ensure they adhere to the journal's format.

Please address these points comprehensively in your revised manuscript and provide a detailed response to each reviewer comment. We look forward to receiving your revised submission.

Best regards,

Reviewer 1 ·

Basic reporting

The Title is more lengthy.
The Abstart miss the phase of the work
The Workflow is not defined properly
The Data provided here is insufficient
The Parameter are nit as per standard.

Experimental design

The diagnosis is very weak
The projection needed more perfections
Results are nit proper
The Methodology is not clear

Validity of the findings

The datasets are unknown as its sources are not clear

The results in the sections are limited
Needed more accurate analysis

Additional comments

English is very weak
Proofread is not done properly

Cite this review as

Reviewer 2 ·

Basic reporting

The tone of online medical services is not described in the abstract.
Why did the authors use CNN along with LSTM when LSTM may be more suitable? Do the authors not think this approach may increase the complexity of this work?
The authors provided sentiment analysis prediction but the need is missing.
The reference style is mixing up with the text, hence causing difficulty in reading.
The introduction does not present valuable information. There are no definitions of terms, no background knowledge, no motivations, and no explicit contributions.
There is no relevant literature in the manuscript, no in-depth related work, and no comparison with the proposed work.
The methodology is well-written but needs revisions in the need of all equations.
No symbols are defined at their first appearances.
The formatting of equations is not aligned and presentable.
Results are not well explained and in-depth discussed.
There should be a table that must show the dataset classes features and other relevant insights.
The paper needs major revisions in writing and structure.

Experimental design

Included in basic reporting.

Validity of the findings

Included in basic reporting.

Cite this review as

Reviewer 3 ·

Basic reporting

very theoretical, old concept.

Experimental design

Experiments should be more innovative.

Comparison should be made some recent machine learning algorithms.

Validity of the findings

Not so approachbale.

Very theoretical.

References need serious change like at least have some new references.

Cite this review as

Reviewer 4 ·

Basic reporting

The paper must address the newly identified challenge(s).
In the introduction section, the authors must clearly state that the problem is challenging based on the latest state-of-the-art methods.

There is a lack of critical analysis of existing research based on existing methods such as Decision Tree Classifiers, Neural Networks, Regular Recursive Neural Networks, existing deep learning network model etc, and their performance and limitations. Could add an analysis of the literature review towards the research gaps which includes the tone classification and recognition.

The review paper is poorly structured/organized.
The methodology needs to be further refined, and justified. The enhancements made to the existing algorithms/techniques need to be presented accordingly so that the contribution to the body of knowledge is clear.

Experimental design

Poor experimental design. There is no explanation of the selected evaluation methods for the experiments performed in this research.

The authors need to provide justifications for selecting the baseline methods, such as SVM, KNN, etc., for comparison purposes. They could combine all the results in Figure 8 into one for easy comparison.

Validity of the findings

The results and findings, specifically the performance of the proposed method for tone classification, are not discussed.

Additional comments

Most of the references are old (more than 10 years). The references need to be updated and must adhere to the journal's format.

Overall, the article lacks research focus on the problem being investigated.

Cite this review as

---

## Round 0.2 · accepted · Accept

Dear authors,

I am pleased to inform you that your manuscript titled "Relation Semantic Fusion in Subgraph for Inductive Link Prediction in Knowledge Graphs" has been accepted for publication in [Journal Name]. The reviewers have evaluated the revisions, and I am happy to confirm that all necessary changes have been satisfactorily addressed.

Thank you for your diligent work in revising the manuscript. We are excited to include your research in our upcoming issue and believe it will make a significant contribution to the field.

Best regards,

Reviewer 3 ·

Basic reporting

Overall, the paper is well-structured and provides a clear and concise overview of the research, effectively emphasizing the problem, methodology, results, and significance.

Experimental design

Inside the paper, given that 1DCNN and BiLSTM are computationally intensive, it would be advantageous to recognize any trade-offs between performance and efficiency.

Validity of the findings

The paper offers a clear and concise description of the methodology, detailing the extraction of 68 features using Librosa, along with the application of 1DCNN for local feature extraction and BiLSTM for capturing global sequential features.

Additional comments

Acceptable

Cite this review as